# Adrenal Gland Primary Neuroblastoma in an Adult Patient: A Case Report and Literature Review

**DOI:** 10.3390/medicina59010033

**Published:** 2022-12-23

**Authors:** Teodora Telecan, Iulia Andras, Maria Raluca Bungardean, Diana Muntean, Claudia Militaru, Ion Perciuleac, Andrei Maga, Nicolae Crisan

**Affiliations:** 1Department of Urology, University of Medicine and Pharmacy Iuliu Hatieganu, 400012 Cluj-Napoca, Romania; 2Department of Urology, Municipal Clinical Hospital, 400139 Cluj-Napoca, Romania; 3Department of Pathology, University of Medicine and Pharmacy Iuliu Hatieganu, 400012 Cluj-Napoca, Romania; 4Department of Pathology, Municipal Clinical Hospital, 400139 Cluj-Napoca, Romania; 5Department of Pharmacology, Toxicology and Clinical Pharmacology, University of Medicine and Pharmacy Iuliu Hatieganu, 400012 Cluj-Napoca, Romania

**Keywords:** adrenal gland, 3D laparoscopic adrenalectomy, neuroblastoma

## Abstract

Neuroblastoma (NB) is an undifferentiated malignant tumor of the sympathetic ganglia, occurring in children under 5 years of age. However, it is a rare histology in adult patients, occurring once per every 10 million patients per year. We present the case of a 68-year-old male patient presented to our department for right lumbar pain, asthenia, loss of weight and altered general status. The contrast-enhanced abdominal computer tomography revealed bilateral adrenal tumoral masses of 149 mm and 82 mm on the right and left sides, respectively, with invasion of the surrounding organs. The patient underwent right 3D laparoscopic adrenalectomy and right radical nephrectomy. The pathological result concluded that the excised tumor was a neuroblastoma of the adrenal gland. The patient followed adjuvant oncological treatment; however, due to disease progression, he passed away 22 months after the surgery. To our knowledge, less than 100 cases of adrenal NB in adult patients have been published, the eldest case being diagnosed at 75 years of age; meanwhile, the largest reported tumor measured 200 mm, and was excised through open surgery. Minimally invasive techniques have been limited so far to smaller, organ-confined diseases, thus making the present case the largest adrenal NB removed entirely laparoscopically. Neuroblastoma in the adult population is a rare finding, with worse prognosis compared to pediatric patients. The available literature does not provide enough data for standardized, multimodal management, as the patients are treated following adapted pediatric protocols, thus reinforcing the need for international, multidisciplinary boards for rare tumors.

## 1. Introduction

Neuroblastoma (NB) is an undifferentiated tumor of the peripheric sympathetic nervous system, being most often located in the paravertebral sympathetic ganglia (chest and retroperitoneum), as well as in the medulla of the adrenal gland [1]. It is the most common extracranial malignancy in young children, occurring once per every 7000 live births [2]. The median age at diagnosis is 19 months, while 90% of NB are diagnosed before 5 years of age [3]. Patients are stratified as low-, intermediate- or high-risk, depending on their age at diagnosis, stage, MYCN proto-oncogene status and tumor cell ploidy [4]. Depending on the risk category, treatment options range from active surveillance approaches and radical surgery for low- and intermediate-risk cases—with overall survival at 5 years being as high as 90%—whilst high-risk patients must undergo neo-adjuvant chemotherapy and surgery, followed by myeloablative and radiation therapy. For this category of patients, the 5-year survival rate does not exceed 50% [5,6].

However, in the adult population, it is a rare occurrence, being estimated as 1 in 10 million adults/year [7]. It is considered that adults have worse prognosis than children, and overall survival rates at 5 years for patients older than 20 years are estimated as 36.3% [3]. Moreover, ATRX translocation is more frequent than MYCN proto-oncogene mutation in this subgroup, being present in 11% of cases, as well as ALK (up to 14% in high-risk NB) and TERT rearrangements (23% of cases) [8,9].

We hereby present the case of an adult patient diagnosed incidentally with NB of the adrenal gland, along with a literature review of similar cases.

## 2. Case Presentation

This is the case of a 68-year-old male patient, who was referred to our department in September 2020 for right lumbar pain, asthenia, loss of weight and altered general status. His past medical history revealed hypertension and peripheral venous insufficiency of the lower limbs, associated with a venous ulcer of the right calf.

Upon examination, a mass could be palpated in the right upper quadrant. The abdominal ultrasound showed a parenchymatous tumor at the upper pole of the right kidney with a diameter of approximately 140 mm.

The patient underwent contrast-enhanced, thoraco-abdominal, and pelvic computer tomography (CECT-TAP), which confirmed a 149 × 106 mm tumoral mass with inhomogeneous structure and necrotic content, located in the right suprarenal region and extending into the upper renal pole, while compressing the right hepatic lobe and adhering at the level of the right diaphragmatic crus. The radiological characteristics indicated its adrenal origin, rather than an upper pole renal carcinoma. A similar mass of 82 × 63 mm was identified contralaterally, extending into the posterior gastric wall, spleen, pancreas, and upper left renal pole. No signs of lymph node involvement or distant metastases have been detected (Figure 1).

A full endocrinological evaluation was conducted, showing an ACTH-dependent Cushing Syndrome, with an ectopic secretory pattern (Table 1). Pituitary MRI did not reveal any abnormalities of the gland.

The case was analyzed by a multidisciplinary uro-oncological board and the surgical indication for right adrenalectomy was established based on the size, the radiologic description and hormonal profile. We proposed a 3D laparoscopic approach. The preoperative blood work revealed moderate anemia (hemoglobin 9.8 g/dL) and urinary tract infection, treated with a full course of antibiotics adapted according to the provided antibiogram.

### 2.1. Surgical Technique

The patient was placed in left lateral decubitus, with the surgical table bent at a 45-degree angle. The intervention was performed using a 5 trocar transperitoneal approach (Figure 2) and the Karl Storz 3D laparoscopy tower (Tuttlingen, Germany).

The ascending colon and duodenum were medialized, exposing the inferior vena cava (IVC). The tumoral mass was identified in the right lumbar area, being adherent to the upper pole of the right kidney, diaphragm, and visceral side of the liver (Figure 3), thus adrenalectomy with radical nephrectomy was decided on.

The renal pedicle comprised of 1 artery and 2 veins was identified, clipped, and divided, as well as the central adrenal vein. The circumferential dissection of the surgical specimen was difficult due to the tumoral invasion of the liver and IVC, but uneventful. The surgical specimen was retrieved through a modified Gibson incision. A peritoneal drainage was placed. The operative time was 210 min and was carried out entirely laparoscopically, with minimum blood loss.

The patient was mobilized on the first post operative day (POD). The drainage was removed on the third POD and the patient was discharged on the seventh POD.

### 2.2. Pathological Report

Macroscopically, the tumoral mass measured 155 × 110 × 70 mm and appeared as a grey mass of hard consistency, polylobate, with multiple areas of necrosis identified on the transversal section.

From a microscopical standpoint, a proliferation of small polyhedric cells was observed in standard hematoxylin—eosin stain, with nuclear atypia and up to 3 mitosis per high power field. The intercellular matrix was fibrillar and poorly represented. Tumoral emboli could be identified in lymphatic and venous vessels, as well as perineural infiltration (Figure 4). Approximatively 20% of the tumoral mass was necrotic tissue. The surgical margins were negative.

Further, immunohistochemical stains were carried out in order to rule out possible differential diagnostics (Table 2). Cytogenetic analysis revealed the absence of MYCN and ALK translocation, while ATRX mutation was present. Taking all of these findings into consideration, as well as the high Ki—67 index (over 80%), the pathologist concluded that the tumor was an undifferentiated neuroblastoma with poor histological prognosis according to the International Neuroblastoma Pathological Classification (INPC), considered Stage 3, L2.

### 2.3. Follow-Up

The patient was referred to the oncologist for adjuvant treatment, starting 2 months after the surgery. He received three cycles of carboplatin and etoposide combined therapy, over the course of 4 months. Due to the rarity of the diagnosis in his age group, an artificial intelligence-based decision support tool was employed (Oncompass^TM^ Gmbh., Schindellegi, Switzerland) that indicated pembrolizumab as being suitable in this case, based on the immunologic profile identified at immunohistochemical analysis. The patient underwent three cycles of pembrolizumab (200 mg intravenously, 21 days apart). However, 9 months after the surgery, the CECT-TAP revealed tumoral progression (left adrenal mass) and local recurrence at the level of the right tumoral bed (right hepatic lobe, right psoas muscle), as well as para-aortic and interaortocaval lymph node metastases. Subsequently, the patient received third line palliative treatment, comprised of three cycles of docetaxel. As the disease progressed, fourth line treatment was initiated, including a combination of doxorubicin, cyclophosphamide, and vincristine, between November 2021 and March 2022. Eventually, the patient needed fifth line therapy, consisting of weekly gemcitabine administration; however, he passed away in July 2022, 22 months after the surgery, due to multiple organ insufficiency caused by the metastases.

## 3. Discussions

Neuroblastoma is an endodermal tumor, having the possibility of developing in any region of the body, along the autonomous nervous system plexi [10]. The adrenal gland is the most common location, followed by the paravertebral ganglia. To date, under 100 cases over 10 years of age have been published in the literature, even less if we restrict the searching criteria to the adult population. A summary of the published cases can be found in Table 3.

To our knowledge, the presented case is one of the eldest in current literature. Zhang et al. [23] reported the case of an even older female patient (75-year-old), who underwent laparoscopic adrenal sparing surgery. The patient refused adjuvant therapy and died due to cancer progression three years later. Furthermore, the current report presents the largest neoplasia of this histology managed completely with a minimally-invasive approach. Schalk et al. [15] and Thapa et al. [24] published cases with masses of 170 mm and 200 mm, respectively, but managed in an open manner. Regarding the laparoscopic approach, it has been taken into consideration for smaller, less invasive tumors. The largest adrenalectomy, that was carried out in a totally laparoscopic fashion, was reported by Ramsingh et al. [22], with the tumor measuring 80 mm.

Although most NB are initially diagnosed as non-secreting incidentalomas, some are prone to secreting hormones that alter the hemodynamic balance of the patient, most often catecholamines, thus mimicking a pheochromocytoma. Alternatively, NB can present as the cause of a Conn’s Syndrome. Although atypical, this is considered to be secondary to renal artery invasion or obstruction, due to large tumors that plunge onto the renal pedicle, increasing the renin production and inducing hyperaldosteronism [27]. The endocrinological evaluation revealed, in this case, an ectopic secretion of ACTH, confirmed by the normal aspect of the pituitary gland on MRI, together with an increased level of ACTH. It is a rare occurrence in adrenal NB, explained by the lack of cellular differentiation characterizing this type of histology, and is much rarer when it comes to ganglioneuroblastomas or ganglioneuromas [28].

Pathological diagnosis represents a challenge, even in the era of cytogenetic and molecular testing. MYCN gene amplification is characteristic for neuroblastomas, encountered in 20–25% of pediatric cases and associated with worse prognosis [29]. However, it is suggested that in adult-onset neuroblastoma, MYCN amplification is less common, while other somatic mutations such as ATRX tend to be more frequent [3]. This was consistent with our patient’s phenotype, as well as with other reported cases in the literature [15]. In terms of histological prognostic factors, the proliferation index (Ki-67) is suggested to indicate the oncological outcomes more accurately than MYCN amplification. Patients having an index over 25% register a survival rate of 34% at three years, compared to 76% in cases with Ki-67 below the cut-off value [30].

To date, there is no consensus regarding the treatment protocol in adult NB. Tumors staged L1 (localized and without invasion of vital nearby structures) and L2 (as long as negative surgical margins are assured) have the possibility of being managed through a multimodal scheme, comprised of surgery, radiation and chemotherapy (most frequently adapted from pediatric protocols), associated with radioactive iodine meta-iodobenzylguanidine (I131 MIBG) therapy [31]. According to the European Society of Endocrinology [32], percutaneous biopsy was ruled out, as the tumor was considered amendable for surgical removal, and hence no prior histological confirmation was needed. Additionally, the sensitivity of the procedure ranges between 50% and 70% [33,34], while the complication rate is as high as 11% [34]. Although locally-advanced, totally 3D laparoscopic adrenalectomy was feasible in our center, as the surgical team has over 10 years of experience in complex minimally-invasive procedures. Therefore, the presented case received surgical treatment for the largest tumoral mass, as well as multiline immuno- and chemotherapy.

Since no standardized protocol is available, personalized oncological assessment has been attempted, however it did not prevent the disease progression. Further chemotherapy protocols have been conducted according to Pediatric Neuroblastoma Protocols, including docetaxel, cyclophosphamide, and doxorubicin, with dose adjustment [35]. Similar approaches have been previously attempted by Genc et al. [14], Schalk et al. [15] and Gupta et al. [16]. However, the reported cases presented distant metastases under systemic chemotherapy and succumbed to the disease at 10 and 9 months after radical surgery, respectively, while the latter underwent additional regional radiotherapy. Tumors characterized by rare histology subtypes lack standardized therapeutic approaches, hence in such scenarios, AI-driven decision support tools may have an important role. In the present case, the main therapeutic targets and their alterations have been identified through cytogenetic analysis. Based on the tumoral genotype, the algorithm synthesizes a hypothetic prognostic model that predicts the expected survival if the patient receives certain immunotherapy molecules, thus ranking each agent and assigning the most suitable one. The process is known as digital therapy planning.

The overall survival reported in the literature is heterogenous, due to the scarcity of cases in the adult population. For the full multimodal protocol, the estimated overall survival at 5 years for stage I and II masses is 83%, while for stages III and IV it drops to 28% [31]. In our case, the patient survived 22 months after the initial diagnosis, similar to some reported cases that had refused adjuvant therapy [23,25]. This finding reinforces the need for international, multidisciplinary tumor boards, in order to reach a consensus regarding the best clinical practice in rare cancers and to increase patients’ survival rates.

The main limitation that we encountered in the management of the presented case was the fact that the final diagnosis was made by ruling out other small blue round cell tumors. Other limitations were the lack of access to I131 MIBG imaging and therapy, as well as the lack of widely applied therapeutic protocols. Regarding the review process, few cases have been published, thus making it difficult to draw solid conclusions regarding incidence, treatment protocols—and how they were adapted from the pediatric population—as well as survival rates.

## 4. Conclusions

Adrenal neuroblastoma is a rare occurrence in the adult and elderly population, with heterogenous and atypical clinical presentation, and worse outcome than those diagnosed in early childhood.

## Figures and Tables

**Figure 1 medicina-59-00033-f001:**
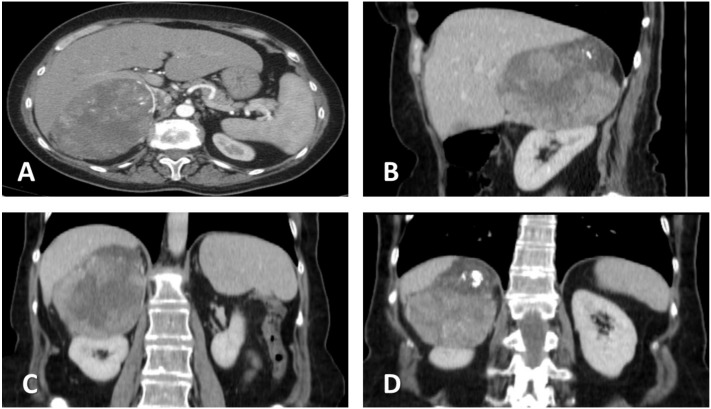
Contrast-enhanced abdominal computer tomography. (**A**) Transversal plane, showing the close contact of the tumor with the right hepatic lobe. (**B**) Sagittal plane. (**C**) Coronal plane, showing possible hepatic and renal invasion, as well as central necrosis. (**D**) Coronal plane, illustrating upper tumoral calcification and suspected invasion of the diaphragm.

**Figure 2 medicina-59-00033-f002:**
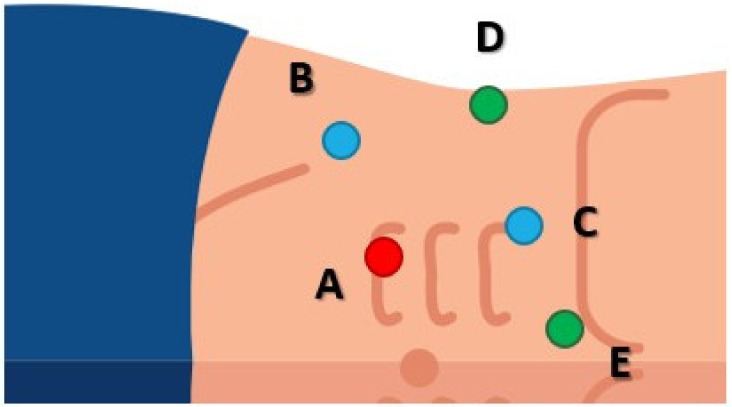
Trocar placement. (**A**) 10 mm optic trocar, placed paraumbilically, at the right border of the rectus abdominis muscle. Two 10 to 12 mm working trocars were inserted for the main surgeon. (**B**) Working trocar for the left hand, placed on the imaginary line drawn between the umbilicus and the anterior iliac spine, 2 cm cranially from the latter. (**C**) Working trocar for the right hand, placed on the line that links the umbilicus with the costal margin, 2 cm below the latter. Next, two trocars of 5 mm each were placed for the assistant surgeon. (**D**) Trocar placed on the mid-axillary line, used for suction. (**E**) Epigastric trocar, used for liver retraction.

**Figure 3 medicina-59-00033-f003:**
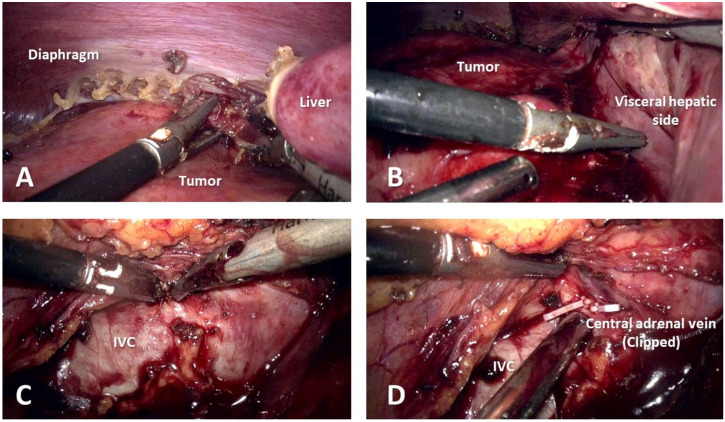
Intraoperative images. (**A**) Tumoral invasion of the diaphragm. (**B**) Infiltration of the visceral side of the liver. (**C**) Difficult isolation of inferior vena cava. (**D**) Central adrenal vein clipped and divided.

**Figure 4 medicina-59-00033-f004:**
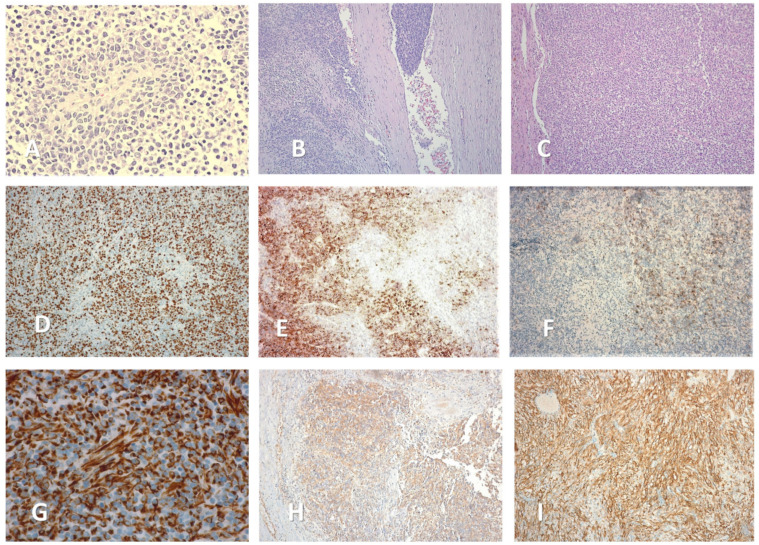
Microscopic aspects of the adrenal mass (magnification size). (**A**) Rosette-like arrangement of tumoral cells, around a blood vessel (×40). (**B**) Venous tumoral emboli (×10). (**C**) Hematoxylin—eosin stain (×10). (**D**) Ki—67 stain (×20). (**E**) Positive CD 10 stain (parcelar pattern) (×10). (**F**) Positive CD 56 stain (parcelar pattern) (×10). (**G**) Positive WT 1 stain (×100). (**H**) Positive Vimentin stain (×10). (**I**) Positive CD 99 stain (×40).

**Table 1 medicina-59-00033-t001:** Hormonal profile, suggestive of ectopic ACTH secretion.

Parameters	Determined Values	Reference Values
24 h urine free cortisol	187.5 µg/24 h	<50 µg/24 h
Serum ACTH	286.2 pg/mL	46–52 pg/mL
Metanephrines	5.7 µmole	<7.1 µmole
Vanillylmandelic acid	38.3 µmole	<50 µmole
Homovanillic acid	46.9 µmole	<82.4 µmole
Aldosterone	14 µg/24 h	3–25 µg/24 h
Seric sodium	139 mEq/L	135–145 mEq/L
Seric potassium	4.4 mEq/L	3.5–5.2 mEq/L

**Table 2 medicina-59-00033-t002:** Immunohistochemical stains, cytogenetic evaluation (*) and differential diagnosis.

Positive Markers	Negative Markers	Excluded Tumors
CD 10 CD 56 CD 99 WT 1 Vimentin ATRX (*)	CD 45 CD 4 CD 8 CD 20 CD 34 CD 31 MelanA Bcl 2 Tdl EMA TTF 1 Chromogranin A NSE S100 PSAP CKAE1/AE3 PAX 8 Inhibin Desmin ALK (*) MYCN (*)	- Melanoma, pulmonary, renal, prostate, germ cells tumors metastases - Adrenal carcinoma - Lymphoma - Neuroendocrine tumors - Primitive Neuroectodermal Tumors (PNET) - Mesonephric carcinoma - Small cells desmoplastic tumors - Sarcoma - Ganglioneuroblastoma

**Table 3 medicina-59-00033-t003:** Cases of adrenal neuroblastoma in adult patients published to date.

No.	Author, Year	Age of the Patient (Years)	Maximum Diameter (mm)	Side	Excreted Hormones	Treatment	Details	Pathological Diagnosis	Complications and Follow-Up
1.	Suzuki et al., 1985 [11]	13	120	Right	Vanilmandelic acid Metanephrines	Open surgery	Right adrenalectomy Lymphadenectomy	Homer-Wright rosettes	Uneventful recovery. Systemic chemotherapy (Vincristine, Doxorubicin, Cyclophosphamide). Alive at 12 months follow-up.
2.	Custodio et al., 1999 [12]	32	140 Tumoral thrombus invading the IVC and extending into the right atrium	Right	None	Open surgery	Right adrenalectomy, nephrectomy and right hepatic lobectomy; IVC thrombectomy and right atriotomy	Synaptophysin (+) Neurofilament (+) No CYT available	Chyloperitoneum and recurrent pleural effusions. Systemic chemotherapy and regional radiotherapy 3 months after surgery. No data regarding disease recurrence available
3.	Genc et al., 2003 [13]	30	66	Left	None	Open surgery	Left adrenalectomy	Homer-Wright rosettes No IHC performed No CYT available	Uneventful recovery. Systemic chemotherapy, disease free at the 12 months follow-up.
4.	Genc et al., 2005 [14]	52	70	Left	Vanilmandelic acid Metanephrines	Open surgery	Left adrenalectomy	No IHC performed No CYT available	Systemic chemotherapy (Cisplatin, Vincristin, Ifosfamide). Multiple liver metastases. Deceased 10 months after the surgery.
5.	Schalk et al., 2005 [15]	51	170 Retroperitoneal, mediastinal and cervical metastases	Right	Cathecolamines	Open surgery	Right adrenalectomy and nephrectomy, and lymphadenectomy. Positive surgical margins	Ki-67 (40%) MYCN (−)	Systemic chemotherapy. Tumoral progression. Deceased 9 months after surgery.
6.	Gupta et al., 2013 [16]	47	N/A	Right	None	Open surgery	Right adrenalectomy and nephrectomy	No IHC performed No CYT available	Tumoral progression after 9 months. Systemic chemotherapy (Cisplatin and Etoposide as first line, then Irinotecan and Carboplatin as second line) and regional radiotherapy (30 Gy/10 fractions/2 weeks)).
7.	Kurokawa et al., 2016 [17]	62	70	Left	None	Open surgery	Left adrenalectomy and splenectomy	NSE (+) Vimentin (+) Synaptophysin (+) S-100 (+) UCHL-1 (−) L26 (−) No CYT available	Tumoral recurrence after 4 months. Radiotherapy (50 Gy).
8.	Casteleti et al., 2017 [18]	11	N/A	Right	None	Open surgery	Right adrenalectomy	MYCN (+)	Recurrence: 5.6 years Survival: 11.37 years Status: alive
		19		Left			Left adrenalectomy		Recurrence: N/A Survival: 3.15 years Status: dead
		11		Left			Left adrenalectomy		Recurrence: 9.3 years Survival: 10 years Status: dead
9.	Manjunath et al., 2018 [19]	15	241 Liver invasion Retroperitoneal adenopaties	Right	Metanephrines	Neoadjuvant chemotherapy (Cisplatin, Etoposide, Doxorubicin, Cyclophosphamide + Open surgery	Right adrenalectomy with upper right renal pole resection and retroperitoneal lymphadenectomy; omental metastasis resection	Horner-Wright rosettes NSE (+) Synaptophysin (+) LCA (−) WT1 (−) CD99 (−)	Uneventful recovery. Regional external radiotherapy. Alive at 6 months follow-up.
10.	Majumder et al., 2018 [20]	23	156 Liver invasion and metastases	Right	None	FNA + Systemic chemotherapy	FNA from one of the liver metastases	Horner-Wright rosettes	Poorly tolerated chemotherapy (Completed 3 cycles).
11.	McCarthy et al., 2019 [21]	11	105 Multiple bone metastases	Left	Catecholamines Chromogranin A	Open surgery	Left adrenalectomy	MYCN (−) ARTX (+)	Systemic chemotherapy (First line: Cyclophosphamide, Topotecan, Doxorubicin, Etoposide, Vincristine, Cisplatin; Second line: Irinotecan/Temozolomide puls Dinutuximab) Deceased 21 months from diagnosis.
		13	67	Left	None	Open surgery	Left adrenalectomy	MYCN (−) ALK (+)	Disease recurrence (left renal fossa mess, para-aortic adenopathy, bone metastases) Systemic chemotherapy (Topotecan, Cytoxan, Cisplatin, Etoposide, Doxorubicin, Vincristine) plus immunotherapy (Dinutuximab) and radiation therapy. Deceased 23 months from diagnosis.
12.	Ramsingh et al., 2019 [22]	22	80	Left	None	Laparoscopic surgery	Left adrenalectomy	Synaptophysin (+) Chromogranin A (+) PGP9.5 (+) CD56 (+) Neurofilament (+) Tyrosine hydroxylase (+) NSE (+) NB84A (+) Ki-67 (10%)	Uneventful recovery. Platinum-based chemotherapy.
13.	Zhang et al., 2019 [23]	75	45	Left	Aldosterone	Laparoscopic surgery	Left adrenal-sparing tumorectomy	CD56 (+) Synaptophysin (+) Vimentin (+) Ki-67 (>30%) CD99 (−) EMA (−) MyoD1 (−) HHF35 (−) Chromogranin A (−) S-100 (−) No CYT available	Patient refused adjuvant oncological treatment. Multiple brain and lung metastases. Deceased 22 months after the surgery.
14.	Thapa et al., 2020 [24]	35	200	Left	None	Open surgery	Left adrenalectomy and nephrectomy, paraaortic lymph node dissection	Homer-Wright rosettes Low Ki-67 No CYT available	Uneventful recovery. Patient refused adjuvant oncological treatment. No signs of recurrence at 69 months after the surgery.
15.	Xu et al., 2021 [25]	40	53	Left	None	Laparoscopic surgery	Left adrenalectomy	Synaptophysin (+) Chromogranin A (+) CD56 (+) CD99 (+) S-100 (partially +) Ki-67 (80%)	Patient refused adjuvant oncological treatment. Multiple lung metastases. Deceased 36 months after the surgery.
16.	Guzman Gomez et al., 2022 [26]	24	80	Right	Cathecolamines	Open surgery	Right adrenalectomy and retroperitoneal lymphadenectomy	N/A	Local recurrence: paravertebral retroperitoneal mass; retrocrural, retrocaval and para-aortic adenopathies. Systemic chemotherapy (Cisplatin, Etoposide).

CYT = cytogenetic analysis; IHC = immunohistochemical stain; Ki-67 = proliferation index; MYCN = avian myelocytomatosis viral oncogene neuroblastoma derived homolog; NSE = neuron-specific enolase; EMA = epithelial membrane antigen; FNA = fine needle aspiration.

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
