# Peer review of "Adrenal Gland Primary Neuroblastoma in an Adult Patient: A Case Report and Literature Review"

_medicina, 2022, doi:10.3390/medicina59010033_

Round 1

Reviewer 1 Report (New Reviewer)

The manuscript presents a case of Neuroblastoma in a 68 year-old patient and the Authors make a literature review of NB in adult. their literature review, however, seems limited to very old patients. Since NB characteristics are very similar in old and yung adults it would be better to include in the literature review all cases or series including patients older than 10 years.

Author Response

1) The manuscript presents a case of Neuroblastoma in a 68 year-old patient and the Authors make a literature review of NB in adult. their literature review, however, seems limited to very old patients. Since NB characteristics are very similar in old and yung adults it would be better to include in the literature review all cases or series including patients older than 10 years.

            We thank the reviewer for his/her insight. The particularity of the case is represented by the age of the patient, as to our knowledge, is the second-oldest case published worldwide. Therefore, the literature review was focused on adult and rather elderly population.

            In response, we enriched Table 2 with additional cases, with ages between 11 and 24 years.

Reviewer 2 Report (New Reviewer)

Adrenal gland primary neuroblastoma in an adult patient: A case report and literature review 

The logical flow of the topics discussed in the abstract is strange and needs reworking, first the background, then a literature review and finally a conclusion. Please combine background and literature review and then move on to methods, results and conclusion (also check the style of other Medicina papers already accepted and align with that, this is what I have found:

Background and ObjectivesMaterials and Methods, results and conclusion

Neuroblastoma is rare in children and extremely rare in adults, so it makes sense that no well-established treatment/ management protocol is in place for adults.

A good introduction to NB, please expand on any differentiating characteristics between adult and paediatric NB, for example, features that lead to a worse prognosis than most paediatric cases.

For confirming diaphragmatic, renal and hepatic involvement did the authors use any other specific technique other than CECT – TAP?

A comprehensive account of the surgical procedure to remove the tumour, thanks.

In figure 4 (histology sections) please insert a scale bar so your readers can determine the scale and changes in scale between each sample to the next. Also, figure 4 provide any images of other positive markers left out including CD99.

Since ATRX translocation is one of the key features identified, please introduce this in the introduction. The authors will know that other than MYCN which is the strongest predictor of poor prognosis and linked to high risk, ALT (linked to ATRX) and TERT rearrangements are also linked to high-risk cases. Please explain this in the introduction.

Please provide any evidence of diagnostics laboratory tests including catecholamines, renal markers and ACTH etc that were mentioned earlier (a table could be used for this).

I mentioned that any management protocols in place for adult NB should be mentioned in the introduction with due reference (for instance the chemotherapy or the AI-based tool mentioned), please familiarise your reader with these tools with due citations. How was the rarity of the diagnosis linked to using AI for treatment stratification? How did this AI system function and what were the input and parameters used to get this output?

Also, the authors mention that the patient was on carboplatin and etoposide for 4 months and then was shifted to AI-recommended pembrolizumab for around 5 months (9 months combined) and was then notified of disease progression, recurrence and metastasis followed by lines 3-5 of treatment. The way this section is written is slightly confusing since it is not clear if the initial 4 and 5-month period of treatment has led to regression since using the term “recurrence” in the next sentence would largely imply some levels of response did take place, please clarify this.

In the discussion, a very large table traverses multiple pahes appears that recaps the rare cases of adult neuroblastoma, although this is rather strange and I don’t know what purpose it is serving. A written recap of some of the characteristics of previous adult NB cases that tally with the current case would have been sufficient. I, however, think that this paper has been already reviewed and this table has been requested by previous reviewers hence it is ok.

Page 9 the highlighted paragraph is interesting and this is what I was referring to in the introduction; the differentiating properties of NB in adults compared to paediatric cases. Again, it would be really beneficial to the reader to read about some of these features in the intro as well as the discussion. 

An interesting point was raised about the oncological outcome of high Ki-67 compared to MYCN.  MYCN is linked to risk which per se is linked to survival, it is interesting that in some cases Ki-67 index can be more informative.

According to the European Society of Endocrinology [25], percutaneous biopsy was ruled out, as the tumor, although locally-advanced, was considered amendable for surgical removal, hence no prior histological confirmation was needed.  Please recap, in minimal words, why this tumour was amenable to surgical removal.

Do the authors have any recommendations for the scientific community?

Author Response

1) Adrenal gland primary neuroblastoma in an adult patient: A case report and literature review

The logical flow of the topics discussed in the abstract is strange and needs reworking, first the background, then a literature review and finally a conclusion. Please combine background and literature review and then move on to methods, results and conclusion (also check the style of other Medicina papers already accepted and align with that, this is what I have found:

Background and Objectives, Materials and Methods, results and conclusion

          We thank the reviewer for his/her comment. We updated the layout of the abstract.

2) Neuroblastoma is rare in children and extremely rare in adults, so it makes sense that no well-established treatment/ management protocol is in place for adults.

A good introduction to NB, please expand on any differentiating characteristics between adult and paediatric NB, for example, features that lead to a worse prognosis than most paediatric cases.

               We thank the reviewer for the suggestion. We added the requested information. Please refer to rows 52 – 56.

3) For confirming diaphragmatic, renal and hepatic involvement did the authors use any other specific technique other than CECT – TAP?

               We appreciate the reviewer’s question. No, in this case, contrast-enhanced computer tomography of the thorax, abdomen and pelvis was considered enough. The imagistic investigation protocol was discussed with our oncologist, in a multidisciplinary uro-oncological board.

4) A comprehensive account of the surgical procedure to remove the tumour, thanks.

               We thank the reviewer for his/her appreciation.

5) In figure 4 (histology sections) please insert a scale bar so your readers can determine the scale and changes in scale between each sample to the next. Also, figure 4 provide any images of other positive markers left out including CD99.

               We appreciate the reviewer’s remark. We added the CD 99 positive stain (Figure 4, panel I) and mentioned in the description the magnification size of each panel.

6) Since ATRX translocation is one of the key features identified, please introduce this in the introduction. The authors will know that other than MYCN which is the strongest predictor of poor prognosis and linked to high risk, ALT (linked to ATRX) and TERT rearrangements are also linked to high-risk cases. Please explain this in the introduction.

               We find this observation interesting. We added the required information in the Introduction (rows 54 – 56).

7) Please provide any evidence of diagnostics laboratory tests including catecholamines, renal markers and ACTH etc that were mentioned earlier (a table could be used for this).

               We thank the reviewer for his/her observation. We added the requested information.

8) I mentioned that any management protocols in place for adult NB should be mentioned in the introduction with due reference (for instance the chemotherapy or the AI-based tool mentioned), please familiarise your reader with these tools with due citations. How was the rarity of the diagnosis linked to using AI for treatment stratification? How did this AI system function and what were the input and parameters used to get this output?

               We appreciate the reviewer’s observation. The standard chemotherapy protocol was detailed in the Discussions section (see rows 218 – 225). In terms of AI decision support tools, their link with the rarity of the diagnosis is represented by the lack of standardized systemic treatment protocols, especially when first line agents fail. As for the AI system algorithm and input parameters, we offered additional details in rows 225 – 232.

9) Also, the authors mention that the patient was on carboplatin and etoposide for 4 months and then was shifted to AI-recommended pembrolizumab for around 5 months (9 months combined) and was then notified of disease progression, recurrence and metastasis followed by lines 3-5 of treatment. The way this section is written is slightly confusing since it is not clear if the initial 4 and 5-month period of treatment has led to regression since using the term “recurrence” in the next sentence would largely imply some levels of response did take place, please clarify this.

               We thank the reviewer for this comment. We rephrased the follow-up paragraph in accordance with the given suggestions. To summarize, we defined recurrence as de novo tumoral growth at the level of the tumoral bed, as the surgical margins were negative. Other pre-existing and newly developed tumoral sites were referred to as “disease progression”.  

10) In the discussion, a very large table traverses multiple pahes appears that recaps the rare cases of adult neuroblastoma, although this is rather strange and I don’t know what purpose it is serving. A written recap of some of the characteristics of previous adult NB cases that tally with the current case would have been sufficient. I, however, think that this paper has been already reviewed and this table has been requested by previous reviewers hence it is ok.

               We thank the reviewer for his/her insight. We believe that by presenting the published cases individually, the heterogeneity of the disease, as well as different therapeutic protocols are better portraited and easier to follow.

11) Page 9 the highlighted paragraph is interesting and this is what I was referring to in the introduction; the differentiating properties of NB in adults compared to paediatric cases. Again, it would be really beneficial to the reader to read about some of these features in the intro as well as the discussion.

               We appreciate the reviewer’s comment. We mentioned the main difference between pediatric and adult cases in the Introduction, rows 52 – 56.

12) An interesting point was raised about the oncological outcome of high Ki-67 compared to MYCN.  MYCN is linked to risk which per se is linked to survival, it is interesting that in some cases Ki-67 index can be more informative.

               We thank the reviewer for the comment.

13) According to the European Society of Endocrinology [25], percutaneous biopsy was ruled out, as the tumor, although locally-advanced, was considered amendable for surgical removal, hence no prior histological confirmation was needed.  Please recap, in minimal words, why this tumour was amenable to surgical removal.

               We acknowledge the reviewer’s insight. The requested information was added accordingly. Please refer to rows 215 – 217.

14) Do the authors have any recommendations for the scientific community?

               We thank the reviewer for this question. Our recommendation would be for an international, multidisciplinary board for rare tumors to be formed, that shall elaborate adequate treatment protocols for rare cancers, or serve as a reference institution that can be consulted for particular cases (rows 238 – 240).

Round 2

Reviewer 2 Report (New Reviewer)

The authors have addressed my comments. 

This manuscript is a resubmission of an earlier submission. The following is a list of the peer review reports and author responses from that submission.

Round 1

Reviewer 1 Report

The authors describe a putative case of neuroblastoma occurring in a 68 year old patient.  While the background on neuroblastoma occurring in adults is well present, and the case presentation is appropriate with nice illustration of the preoperative imaging and intraoperative pictures, the case lacks in my opinion a solid pathologic diagnosis.  The patient clearly has a small round blue cell tumor which appears quite aggressive, but I do not think the authors have made a strong case that this corresponds to neuroblastoma.   Figure 4 A does not show in my opinion Horner-Wright pseudorosettes, which should have neuropil in their core, but actually appears to correspond to peritheliomatous necrosis, with surviving tumor cells around blood vessels.  The lack of chromogranin expression is concerning, and positive stains reported (CD10, CD56, and vimentin) are too unspecific to be able to render a diagnosis.  The WT1 presented in panel 4-G shows cytoplasmic positivity, not nuclear.  The correct diagnosis in this case may require molecular studies or other, as the differential diagnosis of small blue round cell tumors is complex.

Author Response

The authors describe a putative case of neuroblastoma occurring in a 68 year old patient.  While the background on neuroblastoma occurring in adults is well present, and the case presentation is appropriate with nice illustration of the preoperative imaging and intraoperative pictures, the case lacks in my opinion a solid pathologic diagnosis. 

            We thank the reviewer for his/her appreciation.

The patient clearly has a small round blue cell tumor which appears quite aggressive, but I do not think the authors have made a strong case that this corresponds to neuroblastoma.  

Figure 4 A does not show in my opinion Horner-Wright pseudorosettes, which should have neuropil in their core, but actually appears to correspond to peritheliomatous necrosis, with surviving tumor cells around blood vessels. 

            We appreciate the reviewer’s comment. We have replaced Figure 4 A, as well as the description of the fore mentioned image. Please refer to rows 126 – 131.

The lack of chromogranin expression is concerning, and positive stains reported (CD10, CD56, and vimentin) are too unspecific to be able to render a diagnosis. 

            We thank the reviewer for his/her suggestion. According to diagnosis guidelines, CD10 and CD56 should be represented in this kind of tumors in a parcelar pattern. This was indeed identified in our case; however, we chose to include a close-up of the positive immunohistochemical stain. We replaced Figures 4 E and 4 F.  

The WT1 presented in panel 4-G shows cytoplasmic positivity, not nuclear. 

            We find the reviewer’s comment interesting. However, we interpreted this finding as a characteristic of embryonal cells, suggesting the high degree of aggressiveness of the undifferentiated tumor.

The correct diagnosis in this case may require molecular studies or other, as the differential diagnosis of small blue round cell tumors is complex.

            We strongly agree with the reviewer’s comment. However, the present study was limited by the access to cytogenetic diagnosis, thus the final conclusion was drawn by ruling out other small blue round cell tumors. Please refer to rows 213 – 215.

Reviewer 2 Report

The  authors mentioned that there are about 100 cases of adult  neuroblastoma reported in the literature.   It is a challenging condition and it is a primary childhood  cancer and there is currently extensive literature covering the topic however the experience in the adult is limited. 

 although they mentioned that the patient was treated with chemotherapy commonly used in paediatric protocols it was not clear from the authors whether a paediatric oncologist was consulted  for the chemotherapy treatment of this patient's.  I understand that probably the patient had primary surgery because the initial diagnosis was not clear.   in my centre,  where there was a similar situation neuroblastoma was part of the differential diagnosis and a primary trocar biopsy was done from the tumour which subsequently led to have neoadjuvant chemotherapy 1st followed by surgery which was not the case here in this patient which I understand why the approach was done this way because of the unclear diagnosis for a primary adrenal tumour and the risk of seeding  with the biopsy. although it was not mentioned in the paper  and hence primary surgery was performed,  which I think the author probably need to clarify further why the adopted that approach. 

 in addition the authors did not mention whether full skeletal survey was performed to look for bony Mets before surgery was done or not as I think this may have altered their initial approach however there is the of the condition possibly may have led to this pathway of management.

 finally I think the author need to include more recent references especially from the  extensive paediatric   neuroblastoma literature

Author Response

The  authors mentioned that there are about 100 cases of adult  neuroblastoma reported in the literature.   It is a challenging condition and it is a primary childhood  cancer and there is currently extensive literature covering the topic however the experience in the adult is limited. 

            We thank the reviewer for his/her appreciation of our paper.

Although they mentioned that the patient was treated with chemotherapy commonly used in paediatric protocols it was not clear from the authors whether a paediatric oncologist was consulted  for the chemotherapy treatment of this patient's. 

            We thank the reviewer for this suggestion. Due to the rarity of the diagnosis, an artificial-intelligence based decision support tool was employed and Pembrolizumab was indicated as the best option for first-line immunotherapy. As the disease progressed under this line of treatment, further chemotherapy was conducted after consulting the Pediatric Neuroblastoma Guidelines. Please refer to rows 198 – 205.

I understand that probably the patient had primary surgery because the initial diagnosis was not clear.   in my centre,  where there was a similar situation neuroblastoma was part of the differential diagnosis and a primary trocar biopsy was done from the tumour which subsequently led to have neoadjuvant chemotherapy 1st followed by surgery which was not the case here in this patient which I understand why the approach was done this way because of the unclear diagnosis for a primary adrenal tumour and the risk of seeding  with the biopsy. although it was not mentioned in the paper  and hence primary surgery was performed,  which I think the author probably need to clarify further why the adopted that approach. 

            We thank the reviewer for sharing his/her experience. The decision to perform surgery as the first line treatment was analyzed in our department’s multidisciplinary uro-oncological committee, comprised of urology, pathology, medical oncology and radiation therapy consultants. As the initial presumptive diagnostic was that of an adrenocortical carcinoma, the biopsy was not considered suitable, due to its periprocedural risks and moderate to low sensitivity. We added in the ‘Discussion’ section the current recommendations for percutaneous biopsy of the adrenal masses (rows 191 – 197).

In addition the authors did not mention whether full skeletal survey was performed to look for bony Mets before surgery was done or not as I think this may have altered their initial approach however there is the of the condition possibly may have led to this pathway of management.

            We thank the reviewer for his/her comment. The preoperative radiological assessment did not raise the suspicion of distant metastases (bone, liver, lung); therefore, no further investigations were conducted in order to rule out the possibility of tumoral spreading.

Finally I think the author need to include more recent references especially from the  extensive paediatric   neuroblastoma literature

            We thank the reviewer for this comment. We added more information regarding pediatric neuroblastoma in the ‘Introduction’ section (rows 41 – 50).

Round 2

Reviewer 1 Report

The pathologic diagnosis remains uncertain.